# Systematic Review: The Gut Microbiome and Its Potential Clinical Application in Inflammatory Bowel Disease

**DOI:** 10.3390/microorganisms9050977

**Published:** 2021-04-30

**Authors:** Laila Aldars-García, María Chaparro, Javier P. Gisbert

**Affiliations:** 1Hospital Universitario de La Princesa, Instituto de Investigación Sanitaria Princesa (IIS-IP), Universidad Autónoma de Madrid, 28006 Madrid, Spain; laila.alga@gmail.com (L.A.-G.); mariachs2005@gmail.com (M.C.); 2Centro de Investigación Biomédica en Red de Enfermedades Hepáticas y Digestivas (CIBEREHD), 28006 Madrid, Spain

**Keywords:** gut microbiome, inflammatory bowel disease, Crohn’s disease, ulcerative colitis, biomarkers

## Abstract

Inflammatory bowel disease (IBD) is a chronic relapsing–remitting systemic disease of the gastrointestinal tract. It is well established that the gut microbiome has a profound impact on IBD pathogenesis. Our aim was to systematically review the literature on the IBD gut microbiome and its usefulness to provide microbiome-based biomarkers. A systematic search of the online bibliographic database PubMed from inception to August 2020 with screening in accordance with the Preferred Reporting Items for Systematic Reviews and Meta-Analyses (PRISMA) guidelines was conducted. One-hundred and forty-four papers were eligible for inclusion. There was a wide heterogeneity in microbiome analysis methods or experimental design. The IBD intestinal microbiome was generally characterized by reduced species richness and diversity, and lower temporal stability, while changes in the gut microbiome seemed to play a pivotal role in determining the onset of IBD. Multiple studies have identified certain microbial taxa that are enriched or depleted in IBD, including bacteria, fungi, viruses, and archaea. The two main features in this sense are the decrease in beneficial bacteria and the increase in pathogenic bacteria. Significant differences were also present between remission and relapse IBD status. Shifts in gut microbial community composition and abundance have proven to be valuable as diagnostic biomarkers. The gut microbiome plays a major role in IBD, yet studies need to go from casualty to causality. Longitudinal designs including newly diagnosed treatment-naïve patients are needed to provide insights into the role of microbes in the onset of intestinal inflammation. A better understanding of the human gut microbiome could provide innovative targets for diagnosis, prognosis, treatment and even cure of this relevant disease.

## 1. Introduction

The gastrointestinal microbiota comprises a collection of microbial communities, including viruses, bacteria, archaea and fungi, inhabiting the gastrointestinal tract [1]. The constitution and diversity of the microbiota in different sections of the gastrointestinal tract are highly variable and its concentration increases steadily along it, with small numbers in the stomach and very high concentrations in the colon [2,3]. This community has been linked to many diseases, including inflammatory bowel disease (IBD) [4].

IBD encompasses a group of chronic inflammatory bowel pathologies of idiopathic origin that affect millions of people throughout the world; the two most important pathologies covered by this term are Crohn’s disease (CD) and ulcerative colitis (UC) [5]. IBD is not curable and shows a chronic evolution, with alternating periods of exacerbation and remission. This situation entails a high burden on health care systems, which try to provide treatment and to ensure quality of life for these complex patients who often require lifelong medical attention.

The microbiota of the gastrointestinal tract is frequently proposed as one of the key players in the etiopathogenesis of IBD. Studies in animal models and humans have shown that there is a persistent imbalance of the intestinal microbiome (which refers to the gut microbiota and their collective genetic material) related to IBD, with a substantial body of literature providing evidence for the relation of the human gut microbiome and IBD [4,6,7,8,9,10]. Despite all this evidence, it has been difficult to determine whether these changes in the microbiome are the cause of IBD or rather the result of inflammation after IBD onset. The consequence of this relationship between the human gut and microbes is that pharmacological therapies, diet and other interventions targeted to the host will also significantly impact the gut microbiota. Most of the existing studies attempting to determine whether dysbiosis is causative or a consequence of inflammation had certain limitations, such as disparities in methodologic approaches, including different techniques used to analyze the gut microbiome, different sampling sites (stool/mucosa) or site of inflammation, lack of prospective data, small cohort sizes, restricted focus on bacteria, different disease activities and influence of treatment interventions.

We conducted a systematic review to comprehensively collate the body of evidence surrounding the relationship between the gut microbiome and IBD. Our objective was to describe the associations between IBD and dysbiosis and the potential clinical translation of microbiome-based biomarkers.

## 2. Methodology

### 2.1. Search Strategy

An electronic search was conducted using the MEDLINE database via PubMed to identify published articles on the gut microbiome and IBD, from inception to August 2020. The search strings used were:

[(“ulcerative colitis” [MeSH Terms]) OR (“colitis” [All Fields] AND “ulcerative” [All Fields]) OR (“ulcerative colitis” [All Fields]) OR (“crohn disease” [MeSH Terms]) OR (“crohn” [All Fields] AND “disease” [All Fields]) OR (“crohn disease” [All Fields]) OR (“crohn’s disease” [All Fields]) OR (“inflammatory bowel diseases” [MeSH Terms]) OR (“inflammatory bowel diseases” [All Fields])] AND (“microbiome” [All Fields] OR “microbiota” [All Fields]).

Moreover, the reference lists of the included studies were revised to identify further relevant studies.

The work was conducted in accordance with the Preferred Reporting Items for Systematic Reviews and Meta-Analyses statement in Appendix A [11].

### 2.2. Eligibility Criteria

The inclusion criteria were intestinal microbiome studies comparing IBD patients with controls; performed on fecal, intestinal lavage or intestinal tissue samples; focused on human adults; written in English.

Studies were excluded if they reported data on IBD complications or postsurgery (pouchitis, fistulae, among others); studied other conditions in addition to IBD (irritable bowel syndrome, *Clostridium difficile* infection, primary sclerosing cholangitis, among others); were abstracts from conference proceedings, letters to editor, reviews or reported only one patient.

## 3. Results

A total of 5267 records were identified from the PUBMED database. Of 190 papers remaining after screening, 23 did not include controls, 22 included other pathologies and 2 were in silico studies. A total of 143 papers were ultimately included.

### 3.1. Gut Microbiome Studies in IBD: Methodologic Aspects

The main methodologic characteristics of the studies included in this review are summarized in Table 1 (IBD gut microbiome studies using non-next-generation sequencing [NGS] approaches) and Table 2 (IBD gut microbiome studies using NGS approaches).

#### 3.1.1. Study Design

Across the included studies, populations ranged from 2 to 531 patients, many of them with a small sample size that reduced the precision of the estimations. Thus, since many results are limited by sample size, further studies with larger cohorts are desirable to confirm these results and to clarify the significance of the microbiome in the pathogenesis of IBD.

In addition, to date, most published studies in IBD are cross-sectional (121 out of the 143 reviewed studies). However, longitudinal designs are required to capture changes that precede or coincide with disease and symptom onset, and to mechanistically relate microbiome shifts with disease pathogenesis. Overall, longitudinal studies in IBD (only 15% of the included studies) indicate that there is decreased stability in the microbiota composition in UC and CD patients [18,23,25,118,131,132,138,149]. These dynamic changes emphasize the importance of longitudinal sampling for a better understanding of taxa stability in individuals.

The IBD microbiome varies not only over time but also with treatment [80,155,156]. Newly diagnosed patients with no treatment provide an ideal scenario to study the potential etiopathogenesis related to intestinal dysbiosis that occurs in IBD. Mouse and human studies have proven that the gut microbiome is required for disease onset, as germ-free mice rarely develop the disease [157,158], antibiotics can prevent disease onset in mice [159] and ameliorate (but not cure) the disease in humans [160].

However, prior IBD microbiome studies have mostly included subjects with an established treatment; of the 143 microbiome studies included herein, only 11 included treatment-naïve patients [15,66,69,75,81,84,87,93,117,118,123], sometimes only on a small subset of the cohort, and only one was conducted prospectively.

Results on newly diagnosed treatment-naïve patients showed that gut dysbiosis is already established at the beginning of the disease. The dysbiotic profile in the gut of newly diagnosed treatment-naïve IBD patients presents reduced microbial abundance, less biodiversity in the structure of microbial communities, and differential bacterial abundances compared to the profile of established and treated IBD patients or control groups. Conversely, one study showed none or minor microbial differences between these patients and a control group [84].

Current knowledge, despite some controversy, provides valuable insights supporting the idea that microbial alterations may precede IBD onset. Given the limited number of studies in this type of patients, no consistent conclusion can be inferred, and further work is needed to investigate in depth the gut dysbiosis of newly diagnosed treatment-naïve IBD patients.

#### 3.1.2. Microbiome Analysis Methods

Culture-independent and -dependent methods for microbial community analysis have both been used to describe microorganisms from different environments, including the human gut. However, due to the inability to culture the majority of the resident bacteria from the gastrointestinal tract, culture-independent methods have proven much more reliable and faster in profiling complex microbial communities.

Culture-independent techniques are based on sequence divergences of the small subunit ribosomal RNA (16S rRNA) or other target gene regions. Some of these techniques are quantitative real-time PCR (qPCR), denaturing gradient gel electrophoresis (DGGE), terminal restriction fragment length polymorphism (T-RFLP), fluorescence in situ hybridization (FISH), DNA microarrays, and NGS. All these techniques, except for NGS, are referred herein as non-NGS techniques.

Currently, there are many differences in study design and methodology among studies, making translation of basic science results into clinical practice a challenging task. Among the studies included in this review, very few used culture-dependent techniques (7 out of 143); and over the years, NGS became the most employed technique—79 studies used NGS, while 64 studies used non-NGS approaches.

Lately, the most widely used approaches are amplicon gene sequencing, predominantly the 16S rRNA gene (16S rDNA), and whole-genome shotgun sequencing, both NGS techniques.

Another recent technique much less used in this field is flow cytometry. A recent study demonstrated that cytometry fingerprints can be used as a diagnostic tool to classify samples according to CD state [154]. These results highlight the potential of flow cytometry as a tool to conduct rapid and cheaper diagnostics of microbiome-associated diseases.

#### 3.1.3. Sample Type and Site

Currently, bacterial diversity in the human gut is determined through analysis of the luminal content (stool) and mucosal biopsies; however, the stool microbiome differs from the mucosa-associated microbiome [161]. Most of the bacteria are tightly adhered to the mucus and this mucosa-adhered microbiota may be associated with the pathogenesis of the disease [9,76]. Changes observed in stool samples likely represent an indirect measure of what is happening at the mucosal surface, where microorganisms interact more intimately with the host and induce disease.

The studies reviewed herein used fecal data, biopsy data or both, and most of them showed differences between fecal and biopsy samples [13,32,47,80,89,93,96,130], although a few studies found similarities [52,76]. The reported differences in microbial composition related to whether the sample origin was fecal or mucosal indicate that each biological sample represents a different environment thus emphasizing the importance of experimental design. Biopsies are primarily recommended for the dissection of the complex pathogenesis of IBD, whereas feces could effectively detect key biomarkers, enabling non-invasive continuous disease monitoring.

In biopsy samples, sampling site can also be a confounding factor. Many studies have compared the microbiome of inflamed and non-inflamed tissue from the same IBD patient. Regarding the effect of gut inflammation on the microbiota, there are some discrepancies among studies. Some researchers did not find significant differences in the mucosa-associated bacteria between apparently normal and inflamed mucosa in IBD patients [15,66,100,127,128,147]. Conversely, other studies found gut microbiome differences between inflamed and non-inflamed regions in mucosal biopsies [19,44,72,78,120,125,134,144]. This difference was also observed in fungal communities of inflamed mucosa, which are distinguishable from those of the non-inflamed area [63].

In spite of the controversial results, there is evidence supporting that inflamed and non-inflamed tissue samples in both CD and UC may present some differential microbiota composition suggesting that a comparison of mucosal samples obtained from identical sites in IBD patients and non-IBD controls is needed to avoid the confounding effect of inflammation in the assessment of the microbial profile.

#### 3.1.4. Structural and Functional Analysis

IBD microbiome studies have typically focused on characterizing the composition of a community and less attention has been paid to functional profiles of the microbes within a community. Functional information can be inferred from the taxa through bioinformatic approaches or directly assessed via whole-genome shotgun sequencing.

Function is more informative than taxonomy [162] as it provides information on possible mechanisms acting on microbes and on microbe–host interactions, which are important for understanding microbial communities, specially microbiome-related diseases. The loss of a particular function could be more biologically meaningful than the loss of a single or a group of species.

The vast majority of the studies published on the IBD microbiome to date have focused on taxonomy and the reported associations in the IBD gut microbiome are largely limited to identifying high-level taxonomic classification (ranging from phyla to genera) given, for example, the limitations of amplicon gene sequencing for reliable species identification.

Some IBD gut microbiome studies have assessed the change in microbial function compared to healthy subjects. Outcomes of such studies showed a quite distinct change in microbial functions, such as fecal tryptic activity, oxidative response or lipid and glycan metabolism pathways [52,80,83,88,132]. Based on these results, it is necessary to redirect the study of dysbiosis from a purely compositional definition to a definition that includes functional changes of the microbiota.

### 3.2. Dysbiosis in IBD

The microbiome is different among healthy individuals around the globe [163], and the great differences found between the microbiomes of apparently healthy people complicate the definition of a healthy microbiome. Despite this divergence, the vast and diverse microbial gut community lives in relative balance in healthy individuals. Dysbiosis refers to an imbalance in microbial species, which is commonly associated with impaired gut barrier function and inflammatory activity [164]. It encompasses major traits such as loss of beneficial microbes, expansion of pathobionts, and loss of diversity [3] (Figure 1). The following sections will describe the key alterations found in the gut of IBD patients.

#### 3.2.1. Defining the Gut Microbiome in IBD

Although the gastrointestinal tract contains trillions of resident microorganisms that include bacteria, archaea, fungi and viruses, the studies revised herein highlighted that current research on microbiome is mainly focused on bacteria.

##### Bacterial Dysbiosis

It has consistently been shown that there is a disease-dependent restriction of biodiversity and an imbalanced bacterial composition associated with IBD. The abundance of beneficial microorganisms such as *Clostridium* groups IV and XIVa, *Bacteroides*, *Suterella*, *Roseburia*, *Bifidobacterium* species and *Faecalibacterium prausnitzii* is reduced, whereas some pathogens such as Proteobacteria members (including invasive and adherent *Escherichia coli*), *Veillonellaceae*, *Pasteurellaceae*, *Fusobacterium* species, and *Ruminococcus gnavus* are increased [4]. Most of the studies have revealed that in IBD patients, commensal bacteria are depleted and the microbial community is less diverse [14,22,37,48,80,94,106,108,126,143,150,152,153].

The increase in the phylum Proteobacteria, which includes multiple genera considered potentially pathogenic such as *Escherichia*, *Salmonella*, *Yersinia*, *Desulfovibrio*, *Helicobacter* or *Vibrio,* has been extensively reported in IBD patients [17,22,34,35,58,76,113,116,126,135,165].

In the *Firmicutes* phylum, *F. prausnitzii*, an anti-inflammatory commensal bacterium, is frequently decreased in CD, while less evidence has been reported in UC, where it is sometimes increased and in other studies decreased [14,22,51,59,73,89,109,140,142,166,167]. Specific decrease in *Roseburia* spp. in patients with IBD has also been consistently noted [56,59,76,99,115,116,130]. Both bacteria are known to be involved in the production of butyrate, an important energy source for intestinal epithelial cells, which strengthens gut barrier function and exerts important immunomodulatory functions [168]. In this same phylum, the mucin degrader *R. gnavus* is frequently increased in IBD patients’ gut, which may impair barrier stability and contribute to inflammation [38,76,103,111,116,132,140,144].

##### Fungal Dysbiosis

Despite the large body of literature on the IBD gut bacterial microbiome, little has been published on the gut mycobiome; specifically, only nine studies reviewed herein included fungal analysis.

Fungi are ubiquitous and their presence in the gastrointestinal tract has been demonstrated [169]. It was already evidenced many years ago that antibodies directed against mannoproteins of *Saccharomyces cerevisiae* (ASCA) were associated with CD, suggesting an inappropriate immune response to fungi in these patients [170].

Although fungi only constitute approximately 0.1% of the total microbial community in the gut [171], changes in gut mycobiota have been reported in IBD patients. However, results on fungal diversity are controversial; compared to controls, some studies have shown that fungal diversity is decreased in UC patients [107,112], and in CD, diversity and richness have been reported to be either increased [24,63], reduced [103,133], or unchanged [101]. Findings across fungal studies have consistently shown an increase in fungal load, especially in *Candida albicans* [24,63,101,102,107,133].

Nowadays, the exact mechanisms of intestinal fungi in IBD remain unclear and microbiome studies need to include fungi to properly address the complex challenges of this promising field.

##### Viral Dysbiosis

The human gut virome includes a diverse collection of viruses, mostly bacteriophages, directly impacting on human health [172]. In this systematic review, only seven studies included viral analysis [87,90,93,95,129,132,136]. Alterations in IBD gut virome showed an expansion of *Caudovirales* and an inverse correlation between the virome and bacterial microbiome, suggesting an hypothesis where changes in the gut virome may affect bacterial dysbiosis [90,95,129,136]. The use of data on both bacteriome and virome composition would contribute to improve classification between health and disease.

These findings suggest that the loss of virus-bacterium relationships can cause microbiota dysbiosis and intestinal inflammation. However, whether viruses have a direct role in IBD pathogenesis, or merely reflect underlying dysbiosis remains to be determined.

##### Archaeal Dysbiosis

The human gut microbiota also contains prokaryotes of the domain Archaea. Methane-producing archaea (methanogens) have been associated with disorders of the gastrointestinal tract and dysbiosis. Methanogens play an important role in digestion, improving polysaccharide fermentation by preventing accumulation of acids, reaction end-products and hydrogen gas [173].

The two reviewed studies including archaeal analysis have shown that the variable prevalence of methanogens in different individuals may play an important role on IBD pathogenesis [61,71]. Lecours et al. showed that the abundance of *Methanosphaera stastmanae* in fecal samples was significantly higher in IBD patients than in healthy subjects. Interestingly, only IBD patients developed a significant anti-*Msp. stadtmanae* immunoglobulin G response, indicating that the composition of archaeal microbiome appears to be an important determinant of the presence or absence of autoimmunity [61].

The other study demonstrated an inverse association between *Methanobrevibacter smithii* load and susceptibility to IBD, which could be extended to IBD patients in remission as they found that *Mbb. smithii* load was markedly higher in healthy subjects compared to IBD patients [71].

Although archaeal diversity in the gastrointestinal tract is far lower than that of bacteria, these microorganisms can also exert inflammatory effects and their consideration in microbiome studies may be crucial for developing optimal diagnostics and prognostics tools.

##### Disease Activity and Severity

Different disease activity and severity have been described among IBD patients with a given clinically defined condition, suggesting that, in the context of microbiome dysfunction, each condition may present different microbial profiles. The reviewed studies showed a clear difference in the gut microbiota associated with different disease activity and severity in IBD patients.

Dysbiosis was evidenced by Tong et al. [83] at remission, where highly preserved microbial groups accurately classified IBD status during disease quiescence, suggesting that microbial dysbiosis in IBD may be an underlying disorder not only associated with active disease. In general, compared to inactive disease, bacterial diversity and richness are reduced in active disease. Studies of intestinal microbiota in active/inactive IBD patients have consistently shown an increase in *F. prausnitzii* and *Clostridiales* in inactive IBD compared to active IBD, and the increase in *Proteobacteria* in active IBD compared to inactive IBD. Besides, *F. prausnitzii* and *R. hominis* display an inverse correlation with disease activity [51,54,56,59,60,68,114,135,137,139,149].

Some studies showed that the genus *Bifidobacterium* is significantly decreased in stool samples during the active phase of CD and UC compared to the remission phase [43,49,68]. On the contrary, biopsies showed a higher abundance of *Bifidobacterium* during active UC, and the proportion of *Bifidobacterium* was significantly higher in biopsies than in the fecal samples in active CD patients [60]. Some controversial results were also found as other researchers did not find a correlation between microbiota and disease actitivity [35,45,50,101,105,138].

Regarding IBD severity, different microbial abundance was detected in both biopsies and fecal samples from patients with more aggressive disease, and gut dysbiosis was not only related to current activity but also to the course of the disease. In biopsies, *Firmicutes* showed a significant decrease and *Proteobacteria* a significant increase in more aggressive CD [135], and *Bifidobacterium* was inversely correlated with IBD severity [54,135,149]. The risk of flare was associated with reduced microbial richness, increased dysbiosis index and higher individualized microbial instability [74,122,132,137,146,153].

This area is still in its infancy and some results are inconsistent between studies. Several studies have evidenced microbiota signatures of disease activity and severity and the likelihood of a flare-up. However, more research is necessary to identify specific microbial taxa.

### 3.3. Gut Microbiome-Based Biomarkers in IBD

Ideal biomarkers should be easy to obtain, easy to determine, non-invasive, cheap, and capable of providing rapid and reproducible results. Non-invasive tests for IBD are already available, including serum antibodies [174,175], imaging-based screenings [176], and fecal biomarkers [177]. However, endoscopy remains the gold standard for IBD diagnosis, as the aforementioned non-invasive tests are limited to active disease and their outcome can be interfered by diseases other than IBD limiting their clinical utility.

As a non-invasive, cost-effective technique, microbiome-based biomarkers might have great potential for early-stage disease detection and disease course prognosis as well as for treatment based on patient stratification. To this end, several attempts have been made to develop indices of dysbiosis based on relative abundances of selected microbial taxa in IBD patients compared to those of a healthy population. In stool samples, a machine learning algorithm using a combination of 50 operational taxonomic units was able to differentiate remission from active CD [178], and the genera *Collinsella* and *Methanobrevibacter* could be used to differentiate between UC and CD [109]. In biopsies, *Faecalibacteria* and *Papillibacter* were indicators of IBD status [98], *F. prausnitzii* and *E. coli* were used for differential diagnosis of CD (ileal/colonic) [30], supervised learning classification models were able to classify IBD at specific intestinal locations [65], and microbiome shifts predicted patient outcome [62,64,132,137,145,154]. In biopsies, stool and blood a dysbiosis score accurately stratified IBD patients [132].

In the previous sections, differential results on the gut microbiome between CD and UC, IBD and healthy subjects or between different disease activities have been described. Such research on the IBD microbiome has evidenced that (1) alterations in the abundance of certain microbial taxa or (2) in the structure of the microbial community, (3) the decreased bacterial richness and/or diversity and (4) the decreased microbial community stability could be used as potential biomarkers in the field.

Nevertheless, due to the high microbiome diversity between individuals, and within the same individual over time, the predictive value of these potential indicators is currently far below the level required for utility in diagnosis, prognosis, or response to treatment. Nonetheless, the increasing number of microbiome studies along with the use of longitudinal approaches pave the way to the refinement of microbiome-based biomarkers as useful disease indicators.

## 4. Concluding Remarks and Future Perspectives

The study of the human microbiome and its involvement in human health is nowadays one of the most active research topics in biomedicine. A simple search for “Microbiota” and “disease” within PubMed Database reveals almost 28,000 hits to date (august 2020). Given the potential clinical application of the microbiome, the number of studies in this field is rapidly increasing. However, some limitations can be found across these studies, including different methodologic approaches, small cohort sizes, different microbiome analysis methods and sample types and sites, main focus on bacteria, and influence of disease activity and treatment interventions. Therefore, these limitations result in variable findings, difficulty to establish comparison between studies and lack of reproducibility of microbiome signatures across studies.

Recent studies based on novel DNA sequencing methods have revealed major differences in microbial taxonomic and functional composition between IBD patients and healthy individuals. The current knowledge guides us to move our focus from community composition to the understanding of the interactions between microbial functions and the IBD gut microbiome.

The microbiota is very specific to an individual and variable in time, and therefore studies need to go from searching for correlation to searching for causation through longitudinal approaches. One important factor that we must keep in mind when studying the microbiome is that it is a “living entity” subject to variability. This variability is even more evident in the IBD microbiome. To better understand the IBD–microbiome connection, we require prospective longitudinal studies, along with following populations with early-onset IBD. The question of whether dysbiosis precedes the development of IBD and sets the inflammatory process, or merely reflects the altered immune and metabolic environment of the inflamed mucosa, remains to be answered. For this reason, it is of paramount importance to study newly diagnosed treatment-naïve patients, where the microbiome can be studied at the beginning of the disease and without the influence of any IBD treatment. Developing unified approaches to the accurate quantitative assessment of the gut microbiome would contribute to comparisons among studies and to its further clinical application.

The main feature in IBD gut dysbiosis is the decrease in beneficial bacteria and the increase in pathogens. Gut microbiome studies are mainly focused on bacteria, yet beyond bacteria, the gut microbiome is composed of other microorganisms such as viruses, fungi or archaea, which play a role in IBD etiology and/or in bacterial population control. In addition, it is currently known that disease activity and severity influence the gut microbiome, thereby affecting the results. IBD can be considered as a “multimicrobial” disease with no single causative microorganism, in which more severe disease is linked to reduced gut microbial diversity, and proliferation or reduction in specific taxa. Therefore, future studies should include the whole community for a deeper understanding of this disease.

The usefulness of the gut microbiome as a tool towards targeted non-invasive biomarkers for IBD has been evaluated by compelling studies. An acceptable biomarker may help in early diagnosis and classification of IBD as well as in the prediction of disease outcome. Overall, IBD clinical management would benefit from the identification of microbiome-based biomarkers, which could provide less invasive assessment tools, enable personalized treatments, and reduce the health care economic burden associated with IBD. Collectively, these microbiome data represent a valuable data source that can be continually mined to identify associations between the microbiome and IBD for a deeper pathophysiological understanding which may promote the development of clinical strategies, including disease prevention, treatment, stratification and assessment of high-risk population.

## Figures and Tables

**Figure 1 microorganisms-09-00977-f001:**
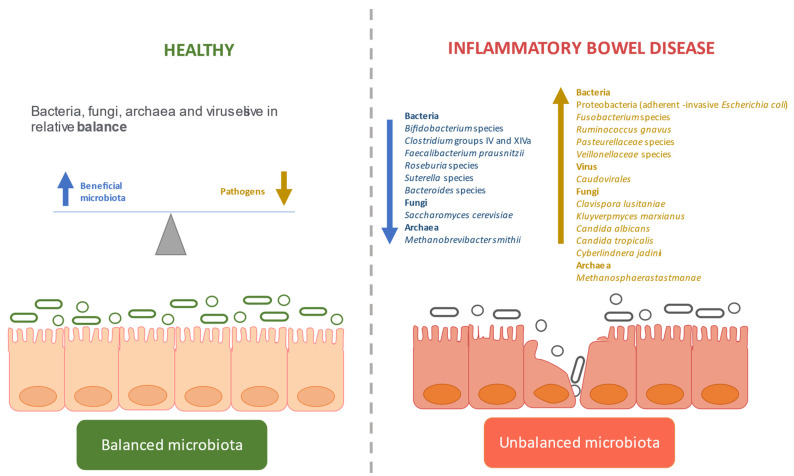
Gut microbiota disturbance in inflammatory bowel disease compared to healthy individuals. Upward arrow indicates increase and downward arrow decrease.

**Table 1 microorganisms-09-00977-t001:** Gut microbiome studies in inflammatory bowel disease using non-next-generation sequencing approaches.

Reference	Year	Treatment	No. of Participants				Disease State	Specimen	Histology	Design	Microbiome Analysis Method	Focus	Microbiota Findings
			CD	UC	IBD/IBDU	HC/C							
Macfarlane et al. [12]	2004	Not naïve	NA	9	NA	10	Active	Biopsy	NA	Cross-sectional	Culture, FISH	Bacteria	UC ▪ Only differences in bifidobacteria were statistically significant. ▪ Peptostreptococci were only present in UC patients.
Lepage et al. [13]	2005	Not naïve	20	11	NA	4	Active/Inactive	Stool and biopsy	Non-inflamed	Cross-sectional	TTGE (16S rDNA V6–V8 region)	Bacteria	CD and UC ▪ Dominant species differ between the mucosa-associated and fecal microbiota. ▪ The microbiota is relatively stable along the distal digestive tract.
Manichanh et al. [14]	2006	Not naïve	6	NA	NA	6	Inactive	Stool	NA	Cross-sectional	Cloning, Sequencing (16S rDNA)	Bacteria	CD ▪ Reduced Firmicutes diversity.
Bibiloni et al. [15]	2006	Naïve	20	15	NA	14	Active	Biopsy	Inflamed/non-inflamed	Cross-sectional	DGGE (16S rDNA V3 region) and qPCR	Bacteria	CD and UC ▪ Bacteria associated with inflamed and non-inflamed tissues did not differ. ▪ UC had more bacteria associated with biopsies than CD. ▪ Bacteroidetes were more prevalent in CD than in UC.
Sokol et al. [16]	2006	Not naïve	NA	9	NA	9	Active	Stool	NA	Cross-sectional	TTGE (16S rDNA V6–V8 region)	Bacteria	UC ▪ Reduced bacterial diversity.
Gophna et al. [17]	2006	Not naïve	6	5	NA	5	Active/Inactive	Biopsy	Inflamed/non-inflamed	Cross-sectional	PCR, cloning, sequencing (16S rDNA)	Bacteria	CD and UC ▪ No significant difference between inflamed and non-inflamed tissues. ▪ In CD, increased *Proteobacteria* and *Bacteroidetes* and reduced *Clostridia.* ▪ No difference between UC and HC.
Scanlan et al. [18]	2006	Not naïve	16	NA	NA	6	Active/Inactive	Stool	NA	Longitudinal	DGGE (16S rDNA)	Bacteria	CD ▪ Lower temporal bacterial stability but higher stability for remission patients. ▪ Lactic acid bacteria spp. varied significantly between the CD groups. ▪ Decrease in *Clostridium* and *Bacteroides* spp.
Zhang et al. [19]	2007	Not naïve	NA	24	NA	NA	Active	Biopsy	Inflamed/non-inflamed	Cross-sectional	DGGE (16S rDNA V3 region)	Bacteria	UC ▪ Lactobacilli and the *Clostridium leptum* subgroup were significantly different between the inflamed and non-inflamed tissues. They were also affected by UC location.
Sepehri et al. [20]	2007	Not naïve	10	15	NA	16	NA	Biopsy	Inflamed/non-inflamed	Cross-sectional	ARISA, T-RFLP	Bacteria	CD and UC ▪ Differences between inflamed and non-inflamed tissues were found. ▪ The non-inflamed tissues form an intermediate population between HC and inflamed tissue for both CD and UC.
Andoh et al. [21]	2007	Not naïve	NA	44	NA	46	Active/Inactive	Stool	NA	Cross-sectional	T-RFLP (16S rDNA)	Bacteria	UC ▪ Bacterial communities are different between HC and active UC patients and between active and inactive patients. ▪ *Eubacterium*, and *Fusobacterium* were predominantly detected in the active patients. ▪ *Lactobacillus* were more predominant in the inactive patients.
Frank et al. [22]	2007	Not naïve	68	61	NA	61	NA	Resected tissue	Inflamed/non-inflamed	Cross-sectional	PCR, cloning, sequencing (16S rDNA)	Bacteria	CD and UC ▪ Significant differences between the microbiotas of CD and UC and those of non-IBD controls. ▪ Depletion of members of the phyla Firmicutes and Bacteroidetes.
Ott et al. [23]	2008	Not naïve	NA	13	NA	5	Active/Inactive	Biopsy	NA	Longitudinal	PCR, cloning and sequencing	Bacteria	UC ▪ Temporal instability and bacterial richness decreased in relapsing patients compared to remission.
Ott et al. [24]	2008	Not naïve	31	26	NA	47	Active	Biopsy	Inflamed	Cross-sectional	DGGE, clone libraries, sequencing, in situ hybridization (18S rDNA)	Fungi	CD ▪ Increased fungal richness and diversity in CD.
Martinez et al. [25]	2008	Not naïve	NA	16	NA	8	Inactive	Stool	NA	Longitudinal	DGGE (16S rDNA V6–V8 region)	Bacteria	UC ▪ Temporal instability and reduced diversity in remission patients.
Dicksved et al. [26]	2008	Not naïve	14	NA	NA	6	Active/Inactive	Stool	NA	Cross-sectional	T-RFLP, cloning and sequencing (16S rDNA)	Bacteria	CD ▪ Decreased bacterial diversity. ▪ Decreased *Bacteroides uniformis* and increased *B. ovatus* and *B. vulgatus.* ▪ Ileal CD bacterial communities were significantly different from HC and colonic CD.
Kuehbacher et al. [27]	2008	Not naïve	42	31	NA	33	Active	Biopsy	Inflamed	Cross-sectional	Clone libraries, sequencing and in situ hybridization (16S rDNA)	Bacteria	CD and UC ▪ TM7 (subgroup of Gram-positive uncultivable bacteria) were more diverse in CD than in UC and non-IBD controls.
Andoh et al. [28]	2008	Not naïve	34	NA	NA	30	Active/Inactive	Stool	NA	Cross-sectional	T-RFLP (16S rDNA)	Bacteria	CD ▪ Decrease in *Clostridium* cluster IV, *Clostridium* cluster XI and subcluster XIVa. ▪ Increase in *Bacteroides* and Enterobacteriales.
Nishikawa et al. [29]	2009	Not naïve	9	NA	NA	11	Active/Inactive	Biopsy	Inflamed/non-inflamed	Longitudinal	T-RFLP (16S rDNA)	Bacteria	UC ▪ Decreased diversity due to loss of commensals. ▪ Decreased diversity in inactive patients compared to active patients.
Willing et al. [30]	2009	Not naïve	14	NA	NA	6	Active/Inactive	Biopsy	Inflamed/non-inflamed	Cross-sectional	T-RFLP, cloning and sequencing, qPCR (16S rDNA)	Bacteria	CD ▪ Ileal CD had a lower abundance of *Faecalibacterium prausnitzii* and an increased abundance of *Escherichia coli* compared to healthy co- twins and colonic CD. ▪ Dysbiosis was significantly correlated to the disease phenotype.
Andoh et al. [31]	2009	Not naïve	NA	2	NA	3Ur	Inactive	Stool	NA	Cross-sectional	T-RFLP (16S rDNA)	Bacteria	UC ▪ Increase in *Clostridium* cluster IX and decreases in *Clostridium* cluster XIVa.
Gillevet et al. [32]	2010	Not naïve	4	2	NA	4	NA	Stool and biopsy	NA	Cross-sectional	LH- PCR, cloning, sequencing, and multitagged pyrosequencing (16S rDNA)	Bacteria	CD and UC ▪ Mucosal microbiome is distinct from the luminal microbiome in HC. ▪ Mucosal microbiome appears to be dysbiotic in IBD.
Rehman et al. [33]	2010	Not naïve	10	10	NA	10	Active	Biopsy	Inflamed	Cross-sectional	PCR, cloning, sequencing (16S rDNA)	Bacteria	CD and UC ▪ Increase in *Escherichia* sp.
Kang et al. [34]	2010	Not naïve	6	NA	NA	6	Inactive	Stool	NA	Cross-sectional	Microarray (16S rDNA)	Bacteria	CD. ▪ Decreased Eubacterium rec- tale, B. fragilis group, B. vulgatus, Ruminococcus albus, R. callidus, R. bromii, and F. prausnitzii. ▪ Increased *Enterococcus* sp., Clostridium difficile, *E. coli*, Shigella flexneri, and *Listeria* sp.
Rowan et al. [35]	2010	Not naïve	NA	20	NA	19	Active/Inactive	Biopsy	NA	Cross-sectional	PCR, qPCR (16S rDNA)	Bacteria	UC ▪ Increase in *Desulfovibrio*, more marked in acute phase.
Andoh et al. [36]	2011	Not naïve	31	31	NA	30	Active/Inactive	Stool	NA	Cross-sectional	T-RFLP (16S rDNA V4–V9)	Bacteria	CD and UC. ▪ Decrease in the *Clostridium* family in active UC and inactive/active CD. ▪ Increase in *Bacteroides.* ▪ Inactive UC tended to be closer to that of HC.
Mondot et al. [37]	2011	Not naïve	16	NA	NA	16	Active	Stool	NA	Cross-sectional	qPCR, RT qPCR (16S rDNA)	Bacteria	CD ▪ Decrease in F. prausnitzii Ruminococcus bromii, Oscillibacter valericigenes, Bifidobacterium bifidum, and E. rectale. ▪ Increase in E. coli and Enterococcus faecium. ▪ More marked increase in E. coli in ileal CD.
Joossens et al. [38]	2011	Not naïve	68	NA	NA	84 Ur + 55	Inactive	Stool	NA	Cross-sectional	DGGE (16S rDNA V3), qPCR	Bacteria	CD ▪ Decrease in Dialister invisus species of Clostridium cluster XIVa, F. prausnitzii and Bifidobacterium adolescentis. ▪ Increase in *R. gnavus*.
Lepage et al. [39]	2011	Not naïve	NA	8	NA	54	Active	Biopsy	NA	Cross-sectional	PCR, cloning, sequencing (16S rDNA)	Bacteria	UC. ▪ Decreased bacterial diversity. ▪ Increase in Actinobacteria and Proteobacteria. ▪ Healthy siblings from discordant twins had more bacteria from the *Lachnospiraceae* and *Ruminococcaceae* families than twins who were both healthy.
Benjamin et al. [40]	2012	Not naïve	103	NA	NA	66	Active	Stool	NA	Cross-sectional	FISH (16S rDNA)	Bacteria	CD ▪ Increase in *Bacteroides*-*Prevotella* in smokers (38.4%) compared with nonsmokers (28.1%). ▪ Increase in bifidobacterial and *Bacteroides-Prevotella*. ▪ Decrease in *F. prausnitzii*.
Hotte et al. [41]	2012	Not naïve	15	14	NA	21	Inactive	Biopsy	Non-inflamed	Cross-sectional	T-RFLP (16S rDNA)	Bacteria	CD and UC ▪ Increase in Proteobacteria compared with HC and UC.
Pistone et al. [42]	2012	Not naïve	35	18	NA	35	NA	Biopsy	Inflamed/non-inflamed	Cross-sectional	PCR	*Mycobacterium avium* subspecies *paratuberculosis*	CD and UC ▪ Increase in *M. avium* subspecies *paratuberculosis* compared to controls.
Andoh et al. [43]	2012	Not naïve	67	NA	NA	121	Active/Inactive	Stool	NA	Longitudinal	T-RFLP (16S rDNA V1–V9)	Bacteria	CD ▪ Decrease in *Clostridia* in active disease and remission and in *Bifidobacterium* in active phase but increased during remission. ▪ Increase in *Bacteroides* genus in active. ▪ Decreased bacterial diversity.
Li et al. [44]	2012	Not naïve	18	NA	NA	9	Active	Stool and biopsy	Inflamed/non-inflamed	Cross-sectional	DGGE (16S rDNA V3 region), sequencing	Bacteria	CD ▪ Decreased bacterial diversity. ▪ Increase in γ-Proteobacteria (especially *E. coli* and *S. flexneri*). ▪ Decrease in reduced Bacteroidetes and Firmicutes. ▪ In ulcerated mucosa, *E. coli* was increased and *F. prausnitzii*, *Lactobacillus coleohominis*, *Bacteroides* sp *and Streptococcus gallolyticus* were decreased compared with the non-ulceated.
Nemoto et al. [45]	2012	Not naïve	NA	48	NA	36	Active/Inactive	Stool	NA	Cross-sectional	Culture, T-RFLP, qPCR	Bacteria	UC ▪ Decreased bacterial diversity. ▪ Decrease in *Bacteroides* and *Clostridium* subcluster XIVab. ▪ Increase in *Enterococcus*.
Vigsnæs et al. [46]	2012	Not naïve	NA	12	NA	6	Active/Inactive	Stool	NA	Cross-sectional	DGGE (16S rDNA, 16S-23S rDNA intergenic spacer region), qPCR	Bacteria	UC. ▪ Different microbiota in active UC compared to HC but in inactive UC compared to HC. ▪ Decrease in *Lactobacillus* spp. and *Akkermansia muciniphila* in active disease.
de Souza et al. [47]	2012	Not naïve	11	7	NA	14	NA	Stool and biopsy	Inflamed/non-inflamed	Cross-sectional	Culture	*E. coli*	CD and UC ▪ Only the mucosa-associated population of *E. coli* was increased, not in stool. The increase was prominent in the ileal CD and rectum and sigmoid of both UC and CD.
Duboc et al. [48]	2013	Not naïve	12	30	NA	26	Active/Inactive	Stool	NA	Cross-sectional	PCR (rDNA), culture	Bacteria	CD and UC ▪ Decrease in the ratio between *F. prausntizii* and *E. coli*
Sha et al. [49]	2013	Not naïve	10	26	NA	14	Active/Inactive	Stool	NA	Cross-sectional	DGGE (16S rDNA V6–V8 region), qPCR	Bacteria	CD and UC. ▪ Decrease in the numbers of Bacteroides–Porphyromonas–Prevotella, Bifidobacterium and B. fragilis in active phase. ▪ Decrease in Helicobacter and Clostridium phylogenetic clusters XI and XIVa in active and inactive phases. ▪ Increase in E. coli in active phases.
Kabeerdoss et al. [50]	2013	Not naïve	20	22	NA	17	Active/Inactive	Stool	NA	Cross-sectional	TTGE (16S rDNA V1–V9), qPCR	*C. leptum* group, *F. prausnitzii*	CD and UC ▪ Decrease in *C. leptum* group bacteria and *F. prausnitzii.* ▪ Decreased bacterial diversity.
Varela et al. [51]	2013	Not naïve	NA	116	NA	29 Ur + 31	Inactive	Stool	NA	Cross-sectional and longitudinal	PCR (16S rDNA), qPCR	*F. prausnitzii*	UC ▪ Decrease in *F. prausnitzii* in patients and relatives. ▪ Recovery of the *F. prausnitzii* population after relapse was associated with remission.
Midtvedt et al. [52]	2013	Not naïve	4	NA	NA	5	Active	Stool and biopsy	Inflamed	Cross-sectional	Microarray	Bacteria	CD ▪ Decrease in Bacteroides in both stool and biopsies.
Fujimoto et al. [53]	2013	Not naïve	47	NA	NA	20	Active/Inactive	Stool	NA	Cross-sectional	qPCR, PCR (16S rDNA V4–V9), T-RFLP	*F. prausnitzii* and *Bilophila wadsworthia*	CD ▪ Decrease in *Clostridia*, including the genus *Faecalibacterium*. ▪ Decreased bacterial diversity.
Fite et al. [54]	2013	Not naïve	NA	33	NA	18	Active	Biopsy	Inflamed	Longitudinal	qPCR	Bacteria	UC ▪ High clinical activity indices were associated with enterobacteria, desulfovibrios, type E *Clostridium perfringens*, and *Enterococcus faecalis*. ▪ Low clinical activity indices were associated with *Clostridium butyricum*, *R. albus*, *Lactobacillus*, *bifidobacterium* and *E. rectale.*
Rajilic-Stojanovic et al. [55]	2013	Not naïve	NA	15	NA	15	Inactive	Stool	NA	Longitudinal	Microarray	Bacteria	UC ▪ Decrease in members of the *Clostridium* cluster IV *R. bromii* et rel. *E. rectale* et rel., *Roseburia* sp., and *Akkermansia* sp. ▪ Increase in *Fusobacterium* sp., *Peptostreptococcus* sp., *Helicobacter* sp., *Campylobacter* sp. and *C. difficile*.
Kumari et al. [56]	2013	Not naïve	NA	26	NA	14	Active/Inactive	Stool	NA	Cross-sectional	FISH, flow cytometry, qPCR (16S rDNA)	Bacteria	UC ▪ Decrease in *C. coccoides* and *C. leptum* clusters. ▪ *F. prausnitzii* and *Roseburia intestinalis* were differentially present in patients with different disease activity.
Hedin et al. [57]	2014	Not naïve	22	NA	NA	25 + 21Ur	Inactive	Stool	NA	Cross-sectional	qPCR (16S rDNA)	Bacteria	CD ▪ Siblings shared dysbiosis pattern with patients (lower concentrations of *F. prausnitzii*, Clostridia cluster IV and *Roseburia* spp.).
Lennon et al. [58]	2014	Not naïve	NA	19	NA	34	Active	Biopsy	NA	Cross-sectional	qPCR (16S rDNA)	*Desulfovibrio* species	UC ▪ No significant differences in *Desulfovibrio* sp. were found between cohorts or at each sampling region between the cohorts.
Machiels et al. [59]	2014	Not naïve	NA	127	NA	447	Active/Inactive	Stool	NA	Cross-sectional	PCR (16S rDNA V3 region) DGGE, sequencing, qPCR	Bacteria	UC ▪ Decrease in Roseburia hominis and F. prausnitzii. ▪ R. hominis and F. prausnitzii showed an inverse correlation with disease activity.
Wang et al. [60]	2014	Not naïve	21	34	NA	21	Active/Inactive	Stool and biopsy	Inflamed/non-inflamed	Cross-sectional	qPCR (16S rDNA)	Bacteria	CD and UC ▪ *Bifidobacterium* was increased in biopsies of active UC patients, and higher in the biopsies than in the fecal samples in active CD patients. ▪ *Lactobacillus* group was s increased in biopsies of active CD patients. ▪ *F. prausnitzii* was decreased in both the fecal and biopsy specimens of the active patients.
Blais Lecours et al. [61]	2014	Not naïve	18	11	NA	29	Active/Inactive	Stool	NA	Cross-sectional	qPCR	Archaea and bacteria	CD and UC ▪ Increase in Methanosphaera stadtmanae.
Fukuda et al. [62]	2014	Not naïve	NA	69	NA	80Ur	Active/Inactive	Stool	NA	Cross-sectional	PCR (16S rDNA, V4–V9 region), T-RFLP	Bacteria	UC ▪ Development of a Discriminant Score based on selected OTUs. ▪ Five differential clusters were obtained indicating a strong association between the gut microbiota and UC *.
Li et al. [63]	2014	Not naïve	19	NA	NA	7	Active	Stool and biopsy	Inflamed/non-inflamed	Cross-sectional	DGGE (18S rDNA), cloning, sequencing	Fungi	CD ▪ Increase in fungal richness and diversity in the inflamed mucosa compared with the non-inflamed mucosa. ▪ Increase in *Candida* spp., *Gibberella moniliformis*, *Alternaria brassicicola*, and *Cryptococcus neoformans*. ▪ In stool, increase in fungal diversity and prevalence in *Candida albicans*, *Aspergillus clavatus*, and *C*. *neoformans.*
Andoh et al. [64]	2014	Not naïve	160	NA	NA	121	Active/Inactive	Stool	NA	Longitudinal	T-RFLP (16S rDNA V1–V9)	Bacteria	CD ▪ Decision tree based on selected OTUs, obtaining 9 groups. ▪ Microbiota profiles may differ according to disease activity.
Wisittipanit et al. [65]	2015	Not naïve	101	89	NA	235	Active/Inactive	Biopsy and lumen aspiration	NA	Cross-sectional	LH-PCR (16S rDNA V1–V2 region)	Bacteria	▪ Development of a computational pipeline to characterise the gut microbial communities. ▪ Model could classify IBD from HC at specific locations and based on disease state *.
Kabeerdoss et al. [66]	2015	Naïve and not naïve	28	32	NA	30	NA	Biopsy	Inflamed/non-inflamed	Cross-sectional	RT-qPCR (16S rDNA)	Bacteria	CD and UC ▪ Increase in *Bacteroides* and *Lactobacillus* in UC patients compared with controls or CD. ▪ Increase in *E. coli* in UC compared with controls. ▪ Decrease in *C. coccoides* group and *C. leptum* group in CD compared with controls. ▪ Decrease in Firmicutes to Bacteroidetes ratio in UC and CD. ▪ No differences between inflamed and non-inflamed tissues were found, nor between treated and untreated patients.
Takeshita et al. [67]	2016	Not naïve	NA	48	NA	34	Active/Inactive	Stool	NA	Cross-sectional	RT-qPCR	Bacteria	UC ▪ Decrease bacterial diversity in active phase. ▪ *Fusicatenibacter saccharivorans* was decreased in active patients and increased in quiescent.
Zhang et al. [68]	2017	Not naïve	132	NA	NA	71	Active/Inactive	Stool	NA	Cross-sectional	Culture	Bacteria	CD ▪ Increase in *E. coli* and *Enterococcus* sp. in active phase compared with inactive and controls.
Vrakas et al. [69]	2017	Naïve and not naïve	12	20	NA	20	Active/Inactive	Biopsy	Inflamed	Cross-sectional	RT-qPCR (16S rDNA)	Bacteria	CD and UC ▪ Increased total bacterial DNA concentration levels in active phase compared to the inactive. ▪ Increase in Bacteroides spp. in active and inactive phases. ▪ Decrease in *C. leptum* group (IV), and *F. prausnitzi* in active and inactive phases.
Zamani et al. [70]	2017	Not naïve	NA	35	NA	60	Active	Biopsy	Inflamed	Cross-sectional	Culture, qPCR	Bacteria	UC ▪ No association between *B. fragilis* and UC. ▪ Enterotoxigenic B. *fragilis* was more prevalent in UC patients with diarrhea.
Ghavami et al. [71]	2018	Not naïve	9	45	NA	47	Active/Inactive	Stool	NA	Cross-sectional	PCR, qPCR (16S rDNA)	Bacteria and *Methanobrevibacter smithii* (Archaea)	CD and UC ▪ Decrease in Methanobrevibacter smithii. ▪ More marked increase in Mbb. smithii in remission than in active phase.
Le Baut et al. [72]	2018	Not naïve	262	NA	NA	76	NA	Resected tissue and biopsy	Inflamed/non-inflamed	Cross-sectional	PCR	*Yersinia* Species	CD ▪ Increase in *Yersinia* species.
Al-Bayati et al. [73]	2018	Not naïve	NA	40	NA	40	NA	Biopsy	Inflamed	Cross-sectional	Culture, PCR (16S rDNA)	Bacteria	UC ▪ Decrease in F. prausnitzii, Prevotella, and Peptostreptococcus productus.
Heidarian et al. [74]	2019	Not naïve	7	22	NA	29	Active/Inactive	Stool	NA	Cross-sectional	qPCR	Bacteria	CD and UC ▪ Decrease in Bacteroides, F. prausnitzii, Prevotella spp., and Methanobrevibacterium. ▪ Decrease in Bacteroides spp., F. prausnitzii, and Prevotella spp. in UC patients with disease activity score greater than 4. ▪ Increase in Streptococcus and Haemophilus in the patients who were at flare.
Vatn et al. [75]	2020	Naïve and not naïve	68	84	12	160	Active/Inactive	Stool	NA	Cross-sectional	GA-map™ (16S rDNA V3–V9 region)	Bacteria	CD and UC ▪ Decrease in Firmicutes and *Eubacterium hallii.* ▪ Increase in *Bifidobacterium* spp., *E. hallii*, Actinobacteria and Firmicutes in ulcerative proctitis, compared to extensive colitis. ▪ No association with disease location in CD.

Abbreviations: CD, Crohn’s disease; UC, ulcerative colitis; IBD, inflammatory bowel disease; IBDU, inflammatory bowel disease unclassified; HC, healthy control; C, control; NA, not applicable; FISH, fluorescence in situ hybridization; TTGE, temporal temperature gradient gel electrophoresis; DGGE, denaturing gradient gel electrophoresis; qPCR, quantitative real-time polymerase chain reaction; ARISA, automated ribosomal intergenic spacer analysis; T-RFLP, terminal restriction fragment length polymorphism; Ur, unaffected relatives; LH-PCR, length heterogeneity polymerase chain reaction; OTUs, operational taxonomic unit. * No microorganisms specified.

**Table 2 microorganisms-09-00977-t002:** Gut microbiome studies in inflammatory bowel disease using next-generation sequencing approaches.

Reference	Year	Treatment	No. of Participants				Disease State	Specimen	Histology	Design	Microbiome Analysis Method	Focus	Microbiota Findings
			CD	UC	IBD/IBDU	HC/C							
Willing et al. [76]	2010	Not naïve	29	16	NA	35	Active/Inactive	Stool and biopsy	Non-inflamed	Cross-sectional	16S rDNA sequencing	Bacteria	CD and UC ▪ Ileal CD differed from colonic CD. ▪ In ileal CD, decrease in *Faecalibacterium* and *Roseburia*, and increase in *Enterobacteriaceae* and *Ruminococcus gnavus*.
Rausch et al. [77]	2011	Not naïve	29	NA	NA	18	Inactive	Biopsy	Non-inflamed	Cross-sectional	16S rDNA V1–V2 region sequencing	Bacteria	CD ▪ Decrease in bacterial diversity. ▪ *Prevotella*, *Lactobacillus*, *Coprobacillus*, *Clostridium*, *Faecalibacterium*, and *Stenotrophomonas* were only present in HC.
Walker et al. [78]	2011	Not naïve	6	6	NA	5	Active	Biopsy	Inflamed/non-inflamed	Cross-sectional	16S rDNA V1–V8 region sequencing	Bacteria	CD and UC ▪ Decrease in bacterial diversity. ▪ Decrease in Firmicutes and increase in Bacteroidetes, and in CD only, *Enterobacteriaceae.* ▪ Differences between inflamed and non-inflamed tissues were found.
Erickson et al. [79]	2012	Not naïve	8	NA	NA	4	Active/Inactive	Stool	NA	Cross-sectional	16S rDNA V1–V2 region and WGS	Bacteria	CD ▪ Decrease in bacterial diversity. ▪ Decrease in Firmicutes in ileal CD.
Morgan et al. [80]	2012	Not naïve	121	75	8	27	Active/Inactive	Stool and biopsy	NA	Cross-sectional	16S rDNA V3–V5 region and WGS	Bacteria	CD and UC ▪ Disease status influenced Firmicutes and Enterobacteriaceae abundances.
Ricanek et al. [81]	2012	Naïve	4	NA	NA	1	Active	Biopsy	Inflamed	Cross-sectional	16S rDNA sequencing	Bacteria	CD ▪ Microbiota of Norwegian CD patients was found to be similar to that of CD patients in other countries.
Li et al. [82]	2012	Not naïve	52	58	NA	60	NA	Biopsy	Non-inflamed	Cross-sectional	16S rDNA V1–V3 and V3–V5 regions sequencing and qPCR	Bacteria	CD and UC ▪ Decrease in *C. coccoides-E. rectales* group in ileal CD compared to control non-IBD. ▪ Decrease in *F. prausnitzii* in CD.
Tong et al. [83]	2013	Not naïve	16	16	NA	32	Inactive	Mucosal lavage	Non-inflamed	Cross-sectional	16S rDNA V4 region sequencing	Bacteria	CD and UC ▪ Decrease in Firmicutes and increase in Proteobacteria and Actinobacteria. ▪ Decrease in microbial diversity
Thorkildsen et al. [84]	2013	Naïve	30	33	3	34	Active	Stool	NA	Cross-sectional	16S rDNA (all regions) sequencing	Bacteria	CD and UC ▪ Increase in Escherichia/Shigella in CD. ▪ Decrease in Faecalibacterium in CD compared to both UC and controls.
Prideaux et al. [85]	2013	Not naïve	22	30	NA	29 +6Ur (CD)	NA	Biopsy	Inflamed/non-inflamed	Cross-sectional	Microarray, 16S rDNA V1–V3 region sequencing	Bacteria	CD and UC ▪ Decrease in microbial diversity and in *Faecalibacterium*, *Coprococcus*, *Dorea*, *Roseburia*, and 2 unclassified gener (from *Lachnospiraceae* and *Clostridiales*) in CD. ▪ In UC, diversity was reduced in Chinese subjects. ▪ Actinobacteria was significantly different between the UC groups. ▪ Decrease in *Coprococcus* and *Dorea* genera in UC.
Chiodini et al. [86]	2013	Not naïve	14	NA	NA	6	NA	Resected tissue	NA	Cross-sectional	16S rDNA V3–V6 region sequencing	Bacteria	CD ▪ Separation of the submucosal and mucosal microbiome and existence of a submucosal bacterial population within diseased tissues.
Pérez-Brocal et al. [87]	2013	Naïve and not naïve	11	NA	NA	8	NA	Stool	NA	Cross-sectional	Viral DNA and 16S rDNA V1–V3 region sequencing	Bacteria and viruses	CD ▪ Decreased bacterial and viral diversity. ▪ *Synechococcus phage* S CBS1 and *Retroviridae* family viruses were more represented in CD. ▪ Increase in Proteobacteria and decrease in Tenericutes, the order Bacteroidales and *Collinsella aerofaciens*.
Davenport et al. [88]	2014	Not naïve	13	14	NA	27	NA	Biopsy	Inflamed	Cross-sectional	16S rDNA V4 region sequencing	Bacteria	CD and UC ▪ Decreased bacterial diversity. ▪ No phylum-level significant differences in Firmicutes or Proteobacteria ▪ Bacteroidetes were only increased in CD.
Chen et al. [89]	2014	Not naïve	26	41	NA	21	Active/Inactive	Stool and biopsy	Inflamed/non-inflamed	Cross-sectional	16S rDNA V1–V3 region sequencing	Bacteria	CD and UC ▪ Decrease in Roseburia, Coprococcus, and Ruminococcus. ▪ Increase in Escherichia-Shigella and Enterococcus. ▪ Fecal- and mucosa-associated microbiota were similar between CD and UC and differed from HC.
Wang et al. [90]	2015	Not naïve	6	4	NA	5	NA	Biopsy	NA	Cross-sectional	RNA sequencing	Bacteria and viruses	CD and UC ▪ Increase in bradyrhizobiaceae, enterobacteriaceae, comamonadaceae, and moraxellaceae families. ▪ Human adenovirus and Herpesviridae sequences were predominant in IBD.
Lavelle et al. [91]	2015	Not naïve	NA	9	NA	4	NA	Luminal brush, mucosal biopsy, mucus gel layer	Inflamed/non-inflamed	Cross-sectional	16S rDNA V4 region sequencing	Bacteria	UC ▪ Spatial variation between the luminal and mucosal communities in both cohorts. ▪ Decrease in *Bacteroidaceae* and *Akkermanseaceae.* ▪ Increase in *Clostridiaceae, Peptostreptococcaceae, Enterobacteriaceae, Ruminococcaceae, Bifidobacteriaceae, Actinomycetaceae* and FJ440089, an uncultured member of the *Prevotellaceae* family.
Chiodini et al. [92]	2015	Not naïve	20	NA	NA	15	NA	Biopsy	Inflamed	Cross-sectional	16S rDNA V4 region sequencing	Bacteria	CD ▪ Distinct sub- mucosal microbiome compared to mucosa and/or fecal material. ▪ *Desulfovibrionales* were present within the submucosal tissues. ▪ Increase in Firmicutes in the subjacent submucosa as compared to the parallel mucosal tissue. ▪ Increase in *Propionibac*- terium spp., *Cloacibacterium* spp., *Parasutterella* spp. and *Methylobacterium* spp.
Pérez-Brocal et al. [93]	2015	Naïve and not naïve	20	NA	NA	20	Active	Stool and biopsy	Inflamed/non-inflamed	Cross-sectional	16S rDNA V1–V3 region and viral DNA/RNA sequencing	Bacteria and viruses	CD ▪ Decrease bacterial diversity in all CD groups. ▪ Increased richness and diversity were observed in feces compared with biopsies. ▪ Increase in *Actinobacteria*, *Gammaproteobacteria*, and *Fusobacteria.*
Vidal et al. [94]	2015	Not naïve	13	NA	NA	7	Active/Inactive	Biopsy	Non-inflamed	Cross-sectional	16S rDNA V1–V5 region sequencing	Bacteria	CD ▪ Decrease in *Clostridia* and increase in Bacteroidetes and Proteobacteria. ▪ No detection of *F. prausnitzii.*
Norman et al. [95]	2015	Not naïve	18	42	NA	12	Active/Inactive	Stool	NA	Cross-sectional	VLP DNA sequencing	Viruses	CD and UC ▪ Increase in *Caudovirales* bacteriophages. ▪ It did not appear that expansion and diversification of the enteric virome was secondary to changes in the microbiota.
Eun et al. [96]	2016	Not naïve	35	NA	NA	15	Inactive	Stool and biopsy	NA	Cross-sectional	16S rDNA V1–V3 region sequencing	Bacteria	CD ▪ Decrease in bacterial diversity. ▪ Increase in Proteobacteria was increased in both fecal and mucosal tissues, and Fusobacteria only in tissue samples. ▪ Increase in *Gammaproteobacteria* and *Fusobacteria* in both fecal and mucosal tissue samples in active phase.
Chiodini et al. [97]	2016	Not naïve	20	NA	NA	15	NA	Biopsy	Inflamed	Cross-sectional	16S rDNA V1–V3 region sequencing	Bacteria	CD ▪ Increase in *Sphingomonadaceae, Alicyclobacillaceae, Methylobacteriaceae, Pseudomonadaceae* and *Prevotellaceae* in the submucosa at the advancing disease margin when compared to the superjacent mucosa (translocation).
Rehman et al. [98]	2016	Not naïve	28	30	NA	30	Inactive	Biopsy	NA	Cross-sectional	16S rDNA V1–V2 region sequencing	Bacteria	CD and UC ▪ Proteobacteria decrease in UC compared with CD and HC. ▪ Different microbial pattern based on geographical origin.
Takahashi et al. [99]	2016	Not naïve	68	NA	NA	10	Active/Inactive	Stool	NA	Cross-sectional	qPCR and 16S rDNA V3–V4 region sequencing	Bacteria	CD ▪ Decrease in Bacteroides, Eubacterium, Faecalibacterium and Ruminococcus. ▪ Increase in Actinomyces and Bifidobacterium.
Forbes et al. [100]	2016	Not naïve	15	21	NA	7	NA	Biopsy	Inflamed/non-inflamed	Cross-sectional	16S rDNA V6 region sequencing	Bacteria	CD and UC ▪ No difference between inflamed and non-inflamed tissues were found. There were only differences between the inflamed and non-inflamed mucosa between CD and UC. ▪ Increase in Bacteroidetes and Fusobacteria in inflamed CD mucosa than in inflamed UC mucosa. ▪ Proteobacteria and Firmicutes were more frequently in the inflamed UC mucosa.
Liguori et al. [101]	2016	Not naïve	23	NA	NA	10	Active/Inactive	Biopsy	Inflamed/non-inflamed	Cross-sectional	qPCR (16S or 18S rDNA) 16S rDNA V3–V4 region and ITS2 sequencing	Bacteria and fungi	CD and UC ▪ Decrease in bacterial diversity. ▪ Increase in Proteobacteria and Fusobacteria. ▪ Increase in fungal load in active phase. *Cystofilobasidiaceae* family and *Candida glabrata* species were overrepresented.
Mar et al. [102]	2016	Not naïve	NA	30	NA	13	NA	Stool	NA	Cross-sectional	16S rDNA V3–V4 region and ITS2 sequencing	Bacteria and fungi	UC ▪ Decrease in bacterial diversity. ▪ Decrease in *Bacteroides* and *Prevotella* species and *Alternaria alternata, Aspergillus flavus, Aspergillus cibarius,* and *Candida sojae.* ▪ Increase in *Streptococcus*, *Bifidobacterium*, and *Enterococcus* and *Candida albicans* and *Debaryomyces.*
Hoarau et al. [103]	2016	Not naïve	20	NA	NA	21 + 28Ur	Active/Inactive	Stool	NA	Cross-sectional	16S rDNA V4 region and ITS1 sequencing	Bacteria and fungi	CD ▪ Increase in Serratia marcescens and E. coli, and Candida tropicalis.
Hedin et al. [104]	2016	Not naïve	21	NA	NA	19+17Ur	Inactive	Biopsy	NA	Cross-sectional	16S rDNA V1–V3 region sequencing	Bacteria	CD ▪ Decrease in bacterial diversity. ▪ Decrease in *F. prausnitzii.*
Naftali et al. [105]	2016	Not naïve	31	NA	NA	5	Active/Inactive	Biopsy	Inflamed	Cross-sectional	16S rDNA V1–V3 region sequencing	Bacteria	CD ▪ Difference between ileal CD compared with colonic CD. This separation was unaffected by the biopsy’s location, its inflammatory state or disease state. ▪ *Faecalibacterium* was strongly reduced in ileal CD compared with colonic CD, whereas *Enterobacteriaceae* were more abundant in the former.
Pedamallu et al. [106]	2016	Not naïve	12	NA	NA	12	NA	Resected tissue	NA	Cross-sectional	WGS	Bacteria	CD ▪ Decrease in Bacteroidetes and Clostridia. ▪ Enrichment of enterotoxigenic *Staphylococcus aureus* and an environmental *Mycobacterium* species within deeper layers of the ileum.
Sokol et al. [107]	2016	Not naïve	149	86	NA	38	Active/Inactive	Stool	NA	Cross-sectional	16S rDNA V3–V5 region and ITS2 sequencing	Bacteria and fungi	CD and UC ▪ Increase in Basidiomycota/Ascomycota ratio and C. albicans. ▪ Decrease in Saccharomyces cerevisiae. ▪ Correlations between bacterial and fungal components.
Santoru et al. [108]	2017	Not naïve	50	82	NA	51	Active/Inactive	Stool	NA	Cross-sectional	16S rDNA V3–V4 region sequencing, qPCR	Bacteria	CD and UC ▪ Increase in Firmicutes, Proteobacteria, Verrucomicrobia, and Fusobacteria. ▪ Decrease in Bacteroidetes and Cyanobacteria.
Pascal et al. [109]	2017	Not naïve	34	33	NA	40 + 71Ur	Active/Inactive	Stool	NA	Longitudinal	16S rDNA V4 region sequencing	Bacteria	CD and UC ▪ Dysbiosis was greater in CD than UC, as shown by a more reduced diversity, a less stable microbial community.
Chen et al. [110]	2017	Not naïve	NA	8	NA	8	NA	Stool	NA	Cross-sectional	16S rDNA V3–V4 region sequencing	Bacteria	UC ▪ Decrease in Firmicutes, (Blautia, Clostridium, Coprococcus and Roseburia). ▪ Decreased bacterial diversity.
Hall et al. [111]	2017	Not naïve	9	10	1	12	Active/Inactive	Stool	NA	Longitudinal	WGS	Bacteria	CD and UC ▪ Increase in facultative anaerobes. ▪ Increase in *R. gnavus*, often co-occurring with increased disease activity.
Qiu et al. [112]	2017	Not naïve	NA	14	NA	15	Active	Biopsy	Inflamed	Cross-sectional	18S rDNA sequencing	Fungi	UC ▪ Increase in Wickerhamomyces, unidentified genus of Saccharomycetales, Aspergillus, Sterigmatomyces, and Candida. ▪ Decrease in Exophiala, Alternaria, Emericella, Epicoccum, Acremonium, Trametes, and Penicillium.
Kennedy et al. [113]	2018	Not naïve	37	NA	NA	54	Inactive	Stool	NA	Cross-sectional	16S rDNA V1–V2 region sequencing	Bacteria	CD ▪ Decrease in bacterial diversity. ▪ Decrease in *Ruminococcaceae, Rikenellaceae,* and *Christensenellaceae*. ▪ Increase in *Enterobacteriaceae*.
Ji et al. [114]	2018	Not naïve	51	66	NA	66	Active/Inactive	Stool	NA	Cross-sectional	16S rDNA V4 region sequencing	Bacteria	CD and UC ▪ Results were different between HC and IBD patients and between active and inactive patients.
Imhann et al. [115]	2018	Not naïve	188	107	18	582	Active/Inactive	Stool	NA	Cross-sectional	16S rDNA V4 region sequencing	Bacteria	CD and UC ▪ Colonic CD was different from that of patients with ileal CD, with a decrease in alpha diversity associated with ileal CD. ▪ Decrease in the genus *Roseburia* was associated with higher IBD risk score.
Nishino et al. [116]	2018	Not naïve	26	43	NA	14	Active/Inactive	Mucosal brush	Non-inflamed	Cross-sectional	16S rDNA V3–V4 region sequencing	Bacteria	CD and UC ▪ No significant difference among anatomical sites within individuals. ▪ Increase in Proteobacteria and decrease in Firmicutes and Bacteroidetes in CD. ▪ Greater abundance of *Escherichia, Ruminococcus (R. gnavus), Clostridium, Cetobacterium, Peptostreptococcus* in CD, and the *Faecalibacterium, Blautia, Bifidobacterium, Roseburia* and *Citrobacter* in UC.
Rojas-Feria et al. [117]	2018	Naïve	13	NA	NA	16	Onset	Stool	NA	Cross-sectional	16S rDNA V1–V3 region sequencing	Bacteria	CD ▪ Decrease in bacterial diversity. ▪ Decrease in Firmicutes and an increase in Bacteroidetes.
Schirmer et al. [118]	2018	Naïve and not naïve	30	21	NA	11	Active/Inactive	Stool	NA	Longitudinal	WGS	Bacteria	CD and UC ▪ Decrease in bacterial diversity. ▪ Decrease in Firmicutes and increase in *Enterobacteriaceae*. ▪ Longitudinal profiles showed taxonomic shifts in community composition over time that coincided with changes in disease severity.
Chiodini et al. [119]	2018	Not naïve	20	NA	NA	15	NA	Resected tissue	Inflamed	Cross-sectional	16S rDNA V1–V3 region sequencing	Bacteria	CD ▪ Increase in bacterial richness. ▪ Bacterial translocation, with two bacterial families (*Comamonadaceae* and *Xanthomonadaceae*), having penetrated the mucosal surfaces.
Hirano et al. [120]	2018	Not naïve	NA	14	NA	14	Active	Biopsy	Inflamed/non-inflamed	Cross-sectional	16S rDNA V4 region sequencing	Bacteria	UC ▪ Decrease in bacterial diversity. ▪ Increase in *Cloacibacterium* and the *Tissierellaceae* and decrease in *Neisseria* in inflamed site when compared to the non-inflamed site.
Ma et al. [121]	2018	Not naïve	15	14	NA	13	Active/Inactive	Stool	NA	Cross-sectional	16S rDNA V4 region sequencing	Bacteria	CD and UC ▪ Increase in Proteobacteria. ▪ Decrease in Bacteroidetes in the active CD compared to inactive CD. ▪ Bacteroidetes showed a negative correlation with the CD activity index scores.
Walujkar et al. [122]	2018	Not naïve	NA	12	NA	7	Active	Biopsy	Inflamed	Longitudinal	16S rDNA V4 region sequencing	Bacteria	UC ▪ Increase in bacterial count in active UC. ▪ Increase in *Stenotrophomonas, Parabacteroides, Elizabethkingia, Pseudomonas, Micrococcus, Ochrobactrum* and *Achromobacter* in active UC.
Moen et al. [123]	2018	Naïve	NA	44	NA	35	Onset	Biopsy	Inflamed/non-inflamed	Cross-sectional	16S rDNA V4 region sequencing	Bacteria	UC ▪ No difference in bacterial diversity. ▪ Proteobacteria were higher in the inflamed tissue compared with the non-inflamed.
Laserna-Mendieta et al. [124]	2018	Not naïve	71	58	NA	75	Active/Inactive	Stool	NA	Cross-sectional	16S rDNA V3–V4 region sequencing	Bacteria	CD and UC ▪ Decreased bacterial diversity. ▪ Decrease in Clostridium cluster IV, *Roseburia,* and *F. prausnitzii* only in CD.
Libertucci et al. [125]	2018	Not naïve	43	NA	NA	10	Active/Inactive	Biopsy	Inflamed/non-inflamed	Cross-sectional	16S rDNA V3 region and ITS2 sequencing	Bacteria and fungi	CD ▪ Increase in *Escherichia* and a decrease in *Firmicutes* in inflamed tissue. ▪ Bacterial diversity did not correlate with inflammation.
Moustafa et al. [126]	2018	Not naïve	45	41	NA	146	Active/Inactive	Stool	NA	Cross-sectional	WGS	Bacteria	CD and UC ▪ Decreased bacterial diversity. ▪ Increase in Proteobacteria and decrease in Bacteroidetes and Firmicutes.
O’Brien et al. [127]	2018	Not naïve	24	NA	NA	17	NA	Biopsy	Inflamed/non-inflamed	Cross-sectional	16S rDNA V1–V3 region sequencing	Bacteria	CD ▪ No bacterial imbalance or reduced diversity in CD aphthous ulcers and adjacent mucosa, relative to control biopsies.
Zakrzewski et al. [128]	2019	Not naïve	15	NA	NA	58	Active	Biopsy	Inflamed/non-inflamed	Cross-sectional	16S rDNA V3–V4 region sequencing	Bacteria	▪ Decrease in bacterial diversity and richness. ▪ Decrease in *F. prausnitzii.*
Zuo et al. [129]	2019	Not naïve	NA	91	NA	76	Active/Inactive	Biopsy	Inflamed/non-inflamed	Cross-sectional	VLP and 16S rDNA sequencing	Viruses	UC ▪ Increase in *Caudovirales* bacteriophages, but decrease in mucosa *Caudovirales* diversity, richness and evenness. ▪ Virome correlated with intestinal inflammation. ▪ Increase in *Escherichia* phage and *Enterobacteria* phage.
Altomare et al. [130]	2019	Not naïve	10	4	NA	11	Active/Inactive	Stool and biopsy	Inflamed/non-inflamed	Cross-sectional	16S rDNA V1–V3 region sequencing	Bacteria	CD and UC ▪ Fecal microbiota was more similar to controls than mucosal microbiota. ▪ In the colon district some specific bacterial biomarkers were identified: *Enterobacteriaceae* for IBD stools, *Bacteroides* for IBD biopsies.
Franzosa et al. [131]	2019	Not naïve	68	53	NA	34	Active/Inactive	Stool	NA	Cross-sectional	WGS	Bacteria	CD and UC ▪ Decreased bacterial diversity. ▪ Decrease in Firmicutes and increase in Proteobacteria. ▪ Disease localization did not have a significant effect among CD subjects.
Lloyd-Price et al. [132]	2019	Not naïve	67	38	NA	27	Active/Inactive	Stool and biopsy	NA	Longitudinal	16S rDNA sequencing and WGS	Bacteria and viruses	CD and UC ▪ Increase in facultative anaerobes at the expense of obligate anaerobes. ▪ Periods of disease activity were marked by increases in temporal taxonomic variability.
Imai et al. [133]	2019	Not naïve	20	18	NA	20	Inactive	Stool	NA	Cross-sectional	16S rDNA V3–V4 region and ITS sequencing	Bacteria and fungi	CD and UC ▪ Decrease in bacterial diversity in CD compared to HC and UC. ▪ No difference in fungal diversity. ▪ Increase in *Candida* in CD compared to HC and UC.
Li et al. [134]	2019	Not naïve	106	NA	88	89	NA	Resected tissue	Inflamed/non-inflamed	Longitudinal	16S rDNA V3–V5 region sequencing, qPCR	Bacteria	CD ▪ Proteobacteria was positively associated with ileal CD and more marked in non-inflamed tissue.
Vester-Andersen et al. [135]	2019	Not naïve	58	82	NA	30	Active/Inactive	Stool	NA	Cross-sectional	16S rDNA V3–V4 region sequencing	Bacteria	CD and UC ▪ Decrease in richness, diversity and Firmicutes in active and in aggressive CD. ▪ Increase in Proteobacteria in CD.
Clooney et al. [136]	2019	Not naïve	27	82	NA	61	Active/Inactive	Stool	NA	Longitudinal	Whole-virome analysis and 16S rDNA V3–V4 region sequencing	Bacteria and viruses	CD and UC ▪ No changes in viral richness. ▪ Increase in Caudovirales. ▪ Changes in virome reflected alterations bacteriome.
Braun et al. [137]	2019	Not naïve	45	NA	NA	22	Inactive	Stool	NA	Longitudinal	16S rDNA V4 region sequencing	Bacteria	CD ▪ Decrease in bacterial diversity. ▪ Inactive patients preceding flare showed a decrease in *Christensenellaceae* and S24.7, and increase in *Gemellaceae* compared with those in remission.
Galazzo et al. [138]	2019	Not naïve	57	NA	NA	15	Active/Inactive	Stool	NA	Longitudinal	16S rDNA V4 region sequencing	Bacteria	CD ▪ Decrease in bacterial diversity and richness. ▪ Microbial community structure was less stable over time.
Sun et al. [139]	2019	Not naïve	NA	58	NA	30	Active/Inactive	Stool	NA	Cross-sectional	16S rDNA V3–V4 region sequencing	Bacteria	UC ▪ Decreased bacterial diversity. ▪ Firmicutes and Bacteroidetes, were the most abundant active UC and inactive UC, respectively. ▪ Increase in Proteobacteria and Fusobacteria and decrease in Firmicutes and Bacteroidetes in active UC.
Yilmaz et al. [140]	2019	Not naïve	270	232	NA	573	Active/Inactive	Biopsy	Inflamed/non-inflamed	Longitudinal	16S rDNA V5–V6 region sequencing	Bacteria	CD and UC ▪ Decrease in diversity in CD compared with UC and HC. ▪ Firmicutes were higher than Bacteroidetes in UC compared with CD.
Magro et al. [141]	2019	Not naïve	18	NA	NA	18	Inactive	Stool	NA	Cross-sectional	16S rDNA V3–V4 region sequencing	Bacteria	UC ▪ Decrease in bacterial diversity. ▪ Increase in Proteobacteria and decrease in the Deltaproteobacteria, *Akkermansia, Oscillospira* and *Saccharomyces cerevisiae*.
Zhang et al. [142]	2019	Not naïve	NA	63	NA	30	Active/Inactive	Stool	NA	Cross-sectional	16S rDNA V4 region sequencing	Bacteria	UC ▪ Decrease in Porphyromonadaceae, Rikeneliaceae, and Lachnospiraceae and increase in Enterococcus and Streptococcus.
Alam et al. [143]	2020	Not naïve	9	11	NA	10	NA	Stool	NA	Cross-sectional	16S rDNA V1–V3 region sequencing	Bacteria	CD and UC ▪ Decrease in bacterial diversity. ▪ Increase in Firmicutes *Prevotellaceae* and decrease Bacteroidetes in UC. ▪ Increase in *Prevotellaceae* and decrease in Bacteroidetes in CD.
Ryan et al. [144]	2020	Not naïve	80	50	NA	31	Active/Inactive	Biopsy	Inflamed/non-inflamed	Cross-sectional	16S rDNA V3–V4 region sequencing	Bacteria	CD and UC ▪ Difference in inflamed and non-inflamed colonic segments in both CD and UC. ▪ Inflammatory status did not appear to affect diversity.
Butera et al. [145]	2020	No naïve	NA	88	NA	24	Active	Biopsy	Inflamed/non-inflamed	Cross-sectional	16S rDNA V3–V4 region sequencing	Bacteria	UC ▪ High IL-13mRNA patients are younger at diagnosis and show higher prevalence of extensive colitis than low IL-13mRNA patients. ▪ Increase in *Prevotella* in patients with high IL-13mRNA tissue content and *Sutterella* and *Acidaminococcus* in patients with low IL-13mRNA tissue content.
Boland et al. [146]	2020	No naïve	101	99	15	48	Active/Inactive	Biopsy	NA	Cross-sectional	16S rDNA V4 region sequencing	Bacteria	CD and UC ▪ CD mucosal biopsy who achieved mucosal healing had lower diversity than biopsies from patients with UC or HC. ▪ Diversity was differently related to mucosal healing in CD and UC.
Olaisen et al. [147]	2020	No naïve	51	NA	NA	40	Active/Inactive	Biopsy	Inflamed/non-inflamed	Cross-sectional	16S rDNA V3–V4 region sequencing	Bacteria	CD ▪ Decreased bacterial diversity. ▪ Overrepresentation of *Tyzzerella* 4. ▪ No difference in diversity in inflamed and non-inflamed ileal mucosa.
Shahir et al. [148]	2020	No naïve	125	NA	NA	23	NA	Biopsy	Inflamed/non-inflamed	Cross-sectional	16S rDNA V1–V2 region sequencing	Bacteria	CD ▪ Decreased bacterial diversity. Distinct profile in colon and ileum. ▪ Increase in obligate anaerobes in the ileum, *B. fragilis* was dramatically increased.
Park et al. [149]	2020	No naïve	370	NA	NA	740	Active/Inactive	Stool	NA	Longitudinal	16S rDNA V3–V4 region sequencing	Bacteria	CD ▪ Diversity was more decreased in patients with worse prognosis. ▪ *E. coli* might be causally involved in CD progression.
Clooney et al. [150]	2020	No naïve	303	228	NA	161	Active/Inactive	Stool	NA	Longitudinal	16S rDNA V3–V4 region sequencing	Bacteria	CD and UC ▪ Decrease in bacterial diversity but increase in variability. ▪ Reduced temporal microbiota stability, particularly in patients with changes in disease activity.
Park et al. [151]	2020	No naïve	10	6	NA	9Ur	Inactive	Stool	NA	Cross-sectional	16S rDNA V3–V4 region sequencing	Bacteria	CD and UC ▪ Decrease in bacterial diversity. ▪ Different diversity and identification of differentially abundant taxa in affected IBD relatives.
Lo Sasso et al. [152]	2020	No naïve	41	43	NA	42	Active	Stool	NA	Cross-sectional	16S rDNA V4 region and WGS	Bacteria	CD and UC ▪ Increase in Proteobacteria, Actinobacteria, and Fusobacteria ▪ Decrease in Firmicutes, Bacteroidetes, and Verrucomicrobia.
Borren et al. [153]	2020	No naïve	108	56	NA	NA	Inactive	Stool	NA	Longitudinal	WGS	Bacteria	CD and UC ▪ Increase in Proteobacteria and Fusobacteria and, at the species level, *Lachnospiraceae*_ *bacterium_2_1_58FAA* in relapse. ▪ Potential microbial biomarker to identify proinflammatory state in quiescent IBD that predisposes to clinical relapse.
Rubbens et al. [154]	2020	No naïve	29	NA	NA	66	Inactive	Stool	NA	Cross-sectional	Flow cytometry and 16S rDNA sequencing	Bacteria	CD ▪ Decrease in bacterial diversity. ▪ Potential of flow cytometry to perform rapid diagnostics of microbiome profile.

Abbreviations: CD, Crohn’s disease; UC, ulcerative colitis; IBD, inflammatory bowel disease; IBDU, inflammatory bowel disease unclassified; HC, healthy control; C, control; NA, not applicable; Ur, unaffected relatives; WGS, whole-genome shotgun sequencing; qPCR, quantitative real-time polymerase chain reaction; VLP, virus-like particle; ITS, internal transcribed spacer.

## Data Availability

All data used, generated or analyzed during this study are included in this published article.

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
