# Peer review of "Systematic Review: The Gut Microbiome and Its Potential Clinical Application in Inflammatory Bowel Disease"

_microorganisms, 2021, doi:10.3390/microorganisms9050977_

Round 1

Reviewer 1 Report

This manuscript reviewed the current knowledge and understanding of gut microbiome and IBD. As a review, the authors did a good job in summarizing information and presenting a consistent and concise perspective of the current knowledge in this field. This is a high quality review providing a good summary of previous findings as well as directions for the field to better research methodology and focus area in future studies. Some suggestions are offered below to improve this manuscript.

Line 173-181, the author mentioned the inconsistent findings from different studies related to the inflammation states of the sampling site. It is a good way to summarize previous studies, and it will be more informative if the author can provide some insights into the reasons leading to such inconsistency.

Line 205-206, the author’s suggestion of shifting the research focus from compositional to functional changes is definitely on point. There are no “core” microbiota shared across all individuals, but a shared “microbiome” generating the same set of biological functions. I am wondering if the author, through reviewing the literature, have identified some known difficulties in studying the functional genome, such as genes with shared functions showed little conserved sequences, resulting in the inability to easily identify such changes in the functions.

In the section of microbiome-based biomarkers, the author summarized some previous findings of the increase or decrease of certain bacteria or OTUs as indicators. In the previous section, the author have established the importance of investigating functional changes versus compositional changes. Therefore, as for the search of a meaningful biomarker, it may also be more useful to focus on the functional groups of microbiome, rather than the specific changes in taxonomy.

It is a good point that the author tried to bring more attention to the microorganisms in the gut other than bacteria. One piece of information that is good to have is the relative abundance in terms of population, or the percentage of functions served by fungi, virus, and/or archaea. Such information helps to established the importance of these groups.

Since the stated objective of this review is to “describe the associations between IBD and dysbiosis”, and some of the limitations are the small cohort sizes and influence of treatment interventions, why not include IBD and microbiota studies using animal models? Animal studies can be more controlled both in terms of study subjects and treatment conditions. Please specify the reason for the inclusion of studies in human cohorts only.

Author Response

Thank you very much for your helpful comments; we have included a new column in tables 1 and 2 where we provide more information regarding your suggestions.

We totally agree with the use of functional groups of microbiome, rather than the specific changes in taxonomy in biomarker discovery. However, currently no biomarkers conclusions could be stated in this sense.

Regarding the inclusion of only human studies in the review, we initially considered to include animal models, but as we aimed to summarize the clinical aspects of the microbiome regarding humans, we decided only to include the human perspective. Besides, as we also tried to summarize the search of microbiome biomarkers, we finally included only human studies as animal models are less accurate to reach conclusions regarding human responses to disease.

Thank you very much for your comments and for giving us the opportunity of sending a reviewed version of the manuscript. We have addressed all the concerns following your suggestions as much as possible. We feel that your comments have considerably improved the final version of the manuscript.

Reviewer 2 Report

Systemic review study entitled Systematic review: Gut Microbiome and its Potential Clinical Application in Inflammatory Bowel Disease by Garcia et is a well-written review article which describes the novel aspects of microbiome in intestinal inflammation. Although the article is very timely and informative, the following are recommended to improve the overall impact of the manuscript.

1) Please elaborate on the section on viral dysbiosis. The authors should discuss the role of other important viruses which are known to play a role in IBD. For example, Rotavirus, Norovirus and Astrovirus are known to produce IBD symptoms such as diarrhea and colitis.

2) In Concluding remarks, and future perspectives section needs revision. Please discuss about the future therapy and potential of clinical application, such as using probiotics, or butyrate producing bacteria (Roseburia) or fecal microbiota transplantation (FMT) which may have the potential to restore a healthy microbiome environment which could be serve as future therapy for IBD and help patients to maintain remission.

Author Response

Many thanks for your suggestions. We added in tables 1 and 2 the findings of each study included in the review, where the role of viruses and the associated symptoms are stated.

We did not include in the review interventional studies, as we aimed to summarize the gut microbiome in IBD from descriptive point of view without including therapeutic aspects. For this reason, we did not discuss about therapies and the further clinical application. We appreciate your suggestion and will consider it for further revisions. However, we followed your suggestion and revised the conclusions section.

Thank you very much for your comments and for giving us the opportunity of sending a reviewed version of the manuscript. We have addressed all the concerns following your suggestions as much as possible. We feel that your comments have considerably improved the final version of the manuscript.

Reviewer 3 Report

Aldars-García et al. presented an extensive review about gut microbiome and its potential clinical application in inflammatory bowel disease. The authors analyzed 144 papers, and they found that microbial taxa enriched or depleted in IBD, including bacteria, fungi, viruses, and archaea. The authors concluded that shifts in gut microbial community composition and abundance are diagnostic biomarkers for IBD progression. In addition, there are significant differences between remission and relapse IBD status regarding the microbial community. The authors provided  comprehensive tables at the end of the systemic review including the gut microbiome studies in IBD using NGS.

In general the review is interesting, inclusive, and well designed.

Author Response

Thank you very much for your comments and for giving us the opportunity of sending this manuscript.

Reviewer 4 Report

Article is written according to PRISMA guidelines and title is interesting. Unfortunately, I must cut through the article to find out what microorganisms play a role in the development of Inflammatory Bowel Disease.

Authors should:

  1. Add in Abstract which microorganisms (genera or species) are the most important in Inflammatory Bowel Disease.
  2. Add Figure with a comparison of an increase or a decrease in microbial levels in healthy patients and persons with Inflammatory Bowel Disease.
  3. Add in Table 1 and 2 column with FINDINGS, it is level of which microorganisms had changed and what does it matter. If are in references RR or OR should be also included. Findings should be added for each presented study.
  4. Nothing follows from the Conclusions. A description of everything and nothing, but no specific data on the potential of gut microbiome on Inflammatory Bowel Disease.

Unfortunately, in recent form manuscript is not acceptable for publication, but I hope that Authors will correct it thoroughly.

Author Response

  1. We added a general sentence defining the microbiota changes in IBD in lines 22-23. We did not specify which genera and species are modified to not extend the abstract, which already has reached the word limit. The specific microorganisms enriched/depleted are described in section 3.2.1. Thanks for your helpful suggestion.
  2. We added in tables 1 and 2 the findings of each study included in the review, where the increase or decrease in microbial levels in IBD compared to healthy patients are stated. We finally did not consider including the figure to not increase the length of the paper, as the modification in tables 1 and has resulted in a considerable increase in the manuscript length. However, following your suggestion, we synthetised all the findings for IBD patients in the new column in tables 1 and 2, and included a sentence in the conclusions (lines 389-390) summarizing the main microbiome changes in IBD.
  3. Thank you for your suggestion. We have included the “findings” for each study in a new column in tables 1 and 2.
  4. Thank you for your suggestion, we made some changes in the conclusion.

Thank you very much for your comments and for giving us the opportunity of sending a reviewed version of the manuscript. We have addressed all the concerns following your suggestions as much as possible. We feel that your comments have considerably improved the final version of the manuscript.

Reviewer 5 Report

The role of the microbiota in maintaining human health is one of the most active areas of research today, I agree. The authors clearly and critically approached the issue of the role of the intestinal microbiota in IBD patients. The authors point out the problems of clinical study design. Newly diagnosed patients with IBD provide the best insight into intestinal microbiota dysbiosis, but there are the fewest of these studies. Likewise, while most studies are geared towards the taxonomy of microbial species, the authors point out well that functional studies are also necessary to indicate the interrelationships of microorganisms in the gut and their impact on the clinical picture of IBD. The authors cite another problem, and that is that microbiota analysis studies have  analyzed different samples and suggest which samples and why should be analyzed. Overall, it is clear to everyone that the microbiota or its dysbiosis plays an important role in the development as well as the clinical outcome of IBD, and further studies are needed to show this. One of the important tubes is the establishment of biomarkers as important tools in the diagnosis of these increasingly common diseases in the population. If possible, it would be good for the authors to create some kind of scheme that would combine the facts stated in the paper. In conclusion, I have no comments other than praise and support for the authors.

Author Response

Thank you very much for your comments; we have included new conclusions in lines 386-390 and 406-408 to synthesize the main facts stated in the paper.

Thank you very much for your comments and for giving us the opportunity of sending a reviewed version of the manuscript. We have addressed all the concerns following your suggestions as much as possible. We feel that your comments have considerably improved the final version of the manuscript.

Round 2

Reviewer 4 Report

Authors corrected their article. Unfortunately, they not added Figure with a comparison of an increase or a decrease in microbial levels in healthy patients and persons with Inflammatory Bowel Disease. I suggest add this figure for better presentation of article topic.

Author Response

Thanks for your helpful suggestion. We included a new figure (figure 1 in line 225) where the increase or decrease in gut microbial levels in IBD compared to healthy individuals is summarized.

Thank you very much for your comments and for giving us the opportunity of sending a reviewed version of the manuscript. We feel that your comments have considerably improved the final version of the manuscript.
